# Predictors of infection, symptoms development, and mortality in people with SARS-CoV-2 living in retirement nursing homes

Andrea De Vito[1]*, Vito Fiore[1], Elija Princic[1], Nicholas Geremia[1], Catello Mario Panu Napodano[1], Alberto Augusto Muredda[1], Ivana Maida[1], Giordano Madeddu[1,2], Sergio Babudieri[1]

1 Unit of Infectious Diseases, Department of Medical, Surgical and Experimental Sciences, University of Sassari, Sassari, Italy, 2 Mediterranean Center for Disease Control, University of Sassari, Sassari, Italy

* andreadevitoaho@gmail.com

## Abstract

**Data Availability Statement:** All relevant data are within the paper and its Supporting Information files.

### Introduction

Since the start of the pandemic, millions of people have been infected, with thousands of deaths. Many foci worldwide have been identified in retirement nursing homes, with a high number of deaths. Our study aims were to evaluate the spread of SARS-CoV-2 in the retirement nursing homes, the predictors to develop symptoms, and death.

### Methods and findings

We conducted a retrospective study enrolling all people living in retirement nursing homes (PLRNH), where at least one SARS-CoV-2 infected person was present. Medical and clinical data were collected. Variables were compared with Student's t-test or Pearson chi-square test as appropriate. Uni- and multivariate analyses were conducted to evaluate variables' influence on infection and symptoms development. Cox proportional-hazards model was used to evaluate 30 days mortality predictors, considering death as the dependent variable. We enrolled 382 subjects. The mean age was 81.15±10.97 years, and males were 140(36.7%). At the multivariate analysis, mental disorders, malignancies, and angiotensin II receptor blockers were predictors of SARS-CoV-2 infection while having a neurological syndrome was associated with a lower risk. Only half of the people with SARS-CoV-2 infection developed symptoms. Chronic obstructive pulmonary disease and neurological syndrome were correlated with an increased risk of developing SARS-CoV-2 related symptoms. Fifty-six (21.2%) people with SARS-CoV-2 infection died; of these, 53 died in the first 30 days after the swab's positivity. Significant factors associated with 30-days mortality were male gender, hypokinetic disease, and the presence of fever and dyspnea. Patients' autonomy and early heparin treatment were related to lower mortality risk.

**Funding:** The authors received no specific funding for this work.

**Competing interests:** The authors have declared that no competing interests exist.

## Conclusions

We evidenced factors associated with infection's risk and death in a setting with high mortality such as retirement nursing homes, that should be carefully considered in the management of PLRNH.

## Introduction

A new severe respiratory syndrome was identified in Wuhan [1] (Hubei Province, China) at the end of December 2019. On January 7, a new Coronavirus was detected and called SARS-CoV-2 [2].

The World Health Organization (WHO) declared SARS-CoV-2 disease (COVID-19) as a public health emergency of international concern and characterized the outbreak as a pandemic on March 12, 2020. Since the start of the pandemic, on October 30, 2020, the total number of SARS-CoV-2 infected people is 44,888,869, with 1,178,475 deaths [3].

The common COVID-19 symptoms are fever, cough, and dyspnea, while less common symptoms are fatigue, headache, anosmia, ageusia, cutaneous manifestation, and gastrointestinal symptoms [4–10]. Furthermore, it has been described as a complication, a higher risk of disseminated intravascular coagulation, and venous thromboembolism [11–13]. The severity and mortality of elderly patients with COVID-19 are higher than those of young and middle-aged patients [14]. Several variables seem to be associated with worse outcomes, such as age > 65, pre-existing concurrent cardiological, and cerebrovascular disease [15].

Treatment with low molecular weight heparin (LMWH) in COVID-19 has been recommended by some expert consensus and is associated with a better prognosis in severe COVID-19 patients with markedly elevated D-dimer [16]. Hydroxychloroquine (HCQ) and azithromycin have been widely in the early treatment of COVID-19 patients. Several trials and metanalyses on HCQ and azithromycin showed a lack of mortality reduction, and now more than 30 countries do not recommend their use in clinical practice. However, the efficacy of HCQ is still debated in the literature [17–24].

Our study aimed to evaluate all people living in Italian retirement homes and identify the risk factor for infection occurrence, symptoms development, and death.

## Methods

### Study conduction

We conducted an observational retrospective cohort study, collecting medical records of people living in retirement nursing homes (PLRNH), in Sassari, Italy, where at least one SARS-CoV-2 infected person was present. Our team evaluated all people present in the facilities from April 9, 2020, to May 31, 2020.

The diagnosis was based on real-time PCR on a nasopharyngeal swab. Medical history and clinical data were collected. Laboratory blood tests were performed when appropriate.

Fever was defined as body temperature > 37.5˚C. Patient compliance was defined as the patient's ability to adhere to medical advice on oral and intravenous treatment regimens and respiratory exercise. Autonomy was defined for a score of 5 or 6 [25], according to the Katz Index of Independence in Activities of Daily Living (ADL). Bedridden was defined as a condition causing inability to move or even sit upright for different medical conditions (e.g., stroke, cognitive impairment, fracture).

SARS-COV-2 infection treatments consisted of hydroxychloroquine 200 mg twice daily, azithromycin 500 mg once daily for five days, or LMWH adjusted by body weight, kidney failure, and HAS-BLED score. Treatment was started only in symptomatic patients. HCQ and azithromycin were not administered on patients with a previous diagnosis of cardiovascular disease, in people with G6PDH deficiency, and those who had a previous side effect to one of these drugs. According to the time of enrolment, the treatment prescription was based on national guidelines, and it was adjusted according to the disease's clinical presentation and drug-drug interactions with patients' chronic therapy.

From the medical records, we extracted the demographic data, medical history (including hypertension, chronic heart disease, chronic obstructive pulmonary disease (COPD), chronic renal disease, obesity, malignancy, diabetes, neurological syndromes, and mental disorders), clinical symptoms, and signs at first evaluation, treatment details, complications, and clinical outcome.

Five medical doctors have collected the medical records, Andrea De Vito, Nicholas Geremia, Elija Princic, Alberto A. Muredda, and Catello Panu Napodano, supervised by Professor Ivana Maida, Giordano Madeddu, and Sergio Babudieri.

## Patients definition

Italian nursing homes are accommodations with extreme variability among accepted people. It follows that the population considered in our study included people requiring low-level support, support in some activities, or with high dependence degree. In all nursing homes included in the study, nurses and assistive personnel were present 24/7, whereas primary care doctors, and other specialists, were present only upon request.

## Medical assessment

Patients' history was collected by the available medical records. Medical assessment was conducted with clinical evaluation every three days, and electrocardiography at time zero and day three of treatment, with a portable device. Blood exams were performed according to the physician's request. Nasopharyngeal swabs were repeated in positive patients every week until the first negative result and until the end of the outbreak among people who tested negative.

## Statistical analysis

Before performing the statistical analysis, data distribution was evaluated with the Kolmogorov-Smirnov test. Data were elaborated as numbers on total (percentages), means ± standard deviation.

Continuous variables with parametric distribution were compared with Student's t-test. Categorical variables were evaluated with the Pearson chi-squared test.

Bivariate analysis was conducted to evaluate variables' influence on infection and symptoms development. The statistical significance level was established as $p < 0.05$.

Regarding 30-days mortality, Cox proportional-hazards model was used for multivariate logistic regression. Independent variables resulting with a $p < 0.2$ at bivariate analysis were included in the multivariate logistic regression. Death was considered as a dependent variable. The significance level was defined as $p < 0.05$.

## Ethical issues

This research has been part of the protocol "COVID-19-SS", approved by the Local Ethical Committee of the University Hospital of Cagliari, with the protocol number PG/2020/9411.

## Results

Since the start of the outbreak, in our province, people living in 63 different retirement nursing homes were tested for SARS-CoV-2 for a total of 1946 subjects. Only in five facilities, there was at least one positive nasopharyngeal swab for SARS-CoV-2. Overall, the positive patients were 264, while the negative patients were 118 (Table 1).

Considering the 382 PLRNH included, 140 (36.7%) were male, with a mean age of 81.15 ± 10.97 years (Table 1). The most common comorbidity was hypertension, present in 223 (58.4%) people, followed by neurological syndromes and mental disorders with 185 (48.4%) and 165 (43.2%) cases, respectively.

Comparing people with a positive swab for SARS-CoV-2 with people with a negative swab (Table 1), there was a significant difference in age (81.93 ± 10.11 vs. 79.4 ± 12.57 years, $p$ = 0.037), having diabetes (21.2% vs. 12.7%, $p$ = 0.048), mental disorders (47.7% vs. 33.1%, $p$ = 0.007), neurological syndromes (44.7% vs. 56.8%), and malignancies (9.8% vs. 1.7%, $p$ = 0.005). Mortality was significantly higher in people with SARS-CoV-2 infection (21.2% vs. 11%, $p$ = 0.017).

At the multivariate analysis (S1 Table), people with mental disorders [Odds Ratio (OR) 1.81 (Confidence Interval (CI) 95% 1.13–2.91) $p$-value = 0.013] and cancer [OR 6.37 (95%CI 1.45–28.03] $p$-value = 0.014] had an increased risk of being infected, while having a neurological syndrome [OR 0.61 (95%CI 0.38–0.98] $p$-value = 0.041] was less associated with the infection. Besides, taking angiotensin II receptor blockers (ARBs) as a chronic treatment increased infection risk [OR 2.40 (95%CI 1.04–5.54) $p$-value 0.04] (Fig 1).

**Table 1. Comparison of characteristics of 382 people living in retirement nursing homes, divided into people with (264) and without (118) SARS-CoV-2 infection.**

|  | Total (382) | Positive (264) | Negative (118) | *p-value*[*] |
|---|---|---|---|---|
| Age, year (mean ± SD) | 81.15 ± 10.97 | 81.93 ± 10.11 | 79.4 ± 12.57 | 0.037 |
| **Gender, n(%)** |  |  |  |  |
| Male | 140 (36.7) | 99 (37.5) | 41 (34.7) | 0.606 |
| **Comorbidities, n(%)** |  |  |  |  |
| BMI > 30 | 71 (18.6) | 53 (20.1) | 18 (15.3) | 0.256 |
| Hypertension | 223 (58.4) | 160 (60.6) | 63 (53.4) | 0.186 |
| Diabetes | 71 (18.6) | 56 (21.2) | 15 (12.7) | 0.048 |
| COPD | 71 (18.6) | 54 (20.5) | 17 (14.4) | 0.160 |
| CHD | 128 (33.5) | 96 (36.4) | 32 (27.1) | 0.077 |
| Mental disorders | 165 (43.2) | 126 (47.7) | 39 (33.1) | 0.007 |
| Neurological syndromes | 185 (48.4) | 118 (44.7) | 67 (56.8) | 0.029 |
| Kidney failure | 28 (7.3) | 20 (7.6) | 8 (6.8) | 0.783 |
| Malignancies | 28 (7.3) | 26 (9.8) | 2 (1.7) | 0.005 |
| Compliance | 228 (59.7) | 166 (62.9) | 62 (52.5) | 0.057 |
| Bedridden | 97 (25.4) | 71 (26.9) | 26 (22) | 0.313 |
| **Chronic Treatment, n(%)** |  |  |  |  |
| ARBs | 49 (12.8) | 41 (15.5) | 8 (6.8) | 0.018 |
| ACE inhibitors | 86 (22.5) | 57 (21.6) | 29 (24.6) | 0.519 |
| **Death, n(%)** | 69 (18.1) | 56 (21.2) | 13 (11) | 0.017 |

[*] Calculated with chi-squared test or t-test as appropriate. SD: standard deviation; BMI: body mass index; COPD: chronic obstructive pulmonary disease; CHD: cardiovascular disease; ARBs: Angiotensin II receptor blockers; ACE: angiotensin-converting enzyme.

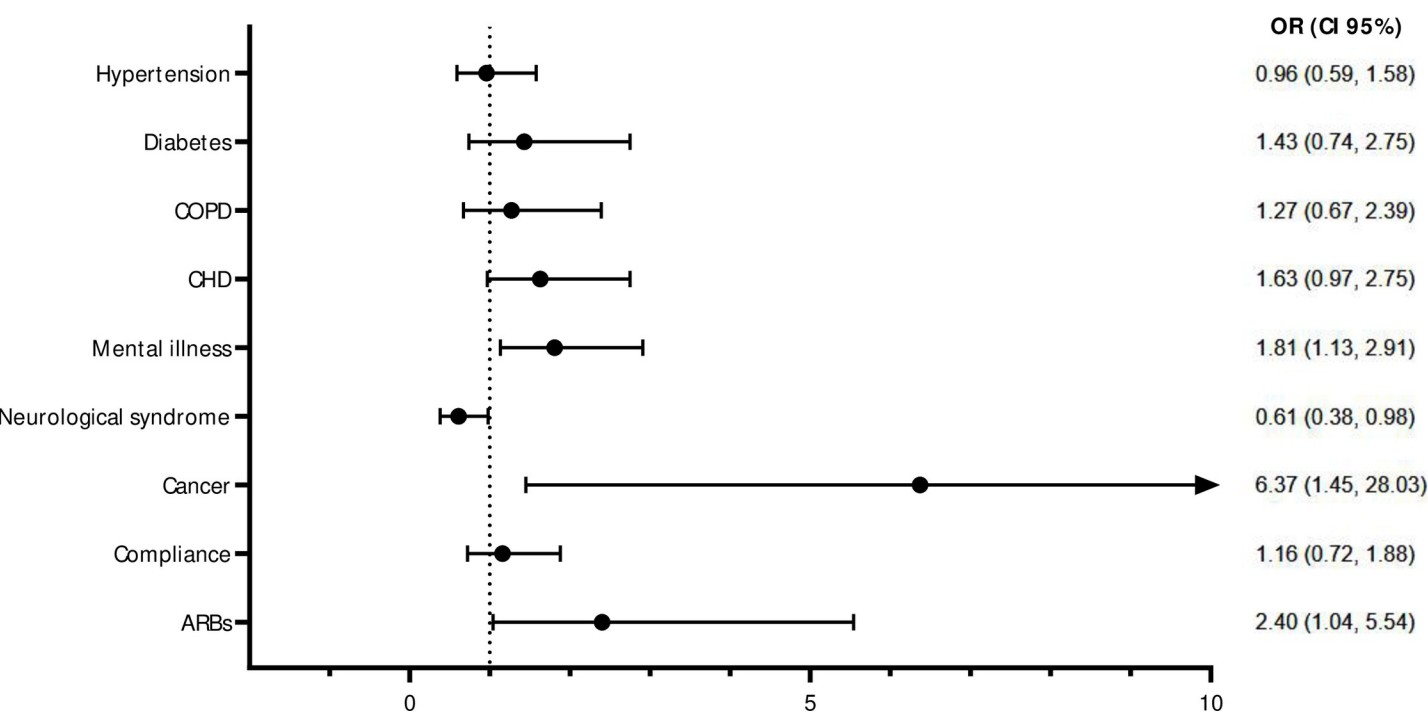

**Fig 1. Forrest Plots of multivariate logistic regression estimates of factors associated with SARS-CoV-2 infection.** Multivariable model includes all variables selected by a backward selection that were retained with a p-value less than 0.2 level. OR: Odds Ratio; CI: Confidence Interval; COPD: chronic obstructive pulmonary disease; CHD: cardiovascular disease; ARBs: Angiotensin II receptor blockers.

### Characteristics of symptomatic patients

Between SARS-CoV-2 infected people 50% developed symptoms (Table 2). The mean age was not significantly different from the asymptomatic subjects (82.3 ± 9.7 vs. 81.5 ± 10.5 years, $p = 0.52$). Significant differences were present in people with COPD (28% vs. 15.2%, $p = 0.011$), CHD (43.9% vs. 31.1%, $p = 0.031$), and neurological syndrome (52.3% vs. 37.1%, $p = 0.013$). Mortality was higher in the symptomatic group (28% vs. 14.4%, $p = 0.007$).

The most common symptoms were fever and dyspnea, present in 90 (68.2%), and 74 (56.1%) patients, respectively. Fifty-five (41.7%) people had both fever and dyspnea. Cough occurred in 27 (20.4%) cases, while 12 (9.1%) complained of severe asthenia.

At the multivariate analysis (S2 Table), having COPD [OR 1.96 (95%CI 1.04–3.70), $p = 0.037$], or a neurological syndrome [OR 1.8 (95%CI 1.07–3.01) $p = .026$] were correlated with an increased risk to develop SARS-CoV-2 related symptoms (S1 Fig).

Among symptomatic people, 124 (93.9%) started SARS-CoV-2 treatment. In particular, 59 (44.7%) received hydroxychloroquine, 48 (36.4%) azithromycin, and 105 (79.5%) heparin. Despite the drugs used, no adverse reactions such as QT prolongation were found.

### Mortality

Fifty-six (21.2%) people with SARS-CoV-2 infection died; of them, 53 died in the first 30 days after the swab's positivity. Dividing people by age, mortality was higher in people with 80 years and above (24.6%), while it was significantly lower in the other age ranges (Fig 2).

The deceased's mean age was significantly higher than the survivors' mean age (85.3 ± 6.3 vs. 81 ± 10.7 years, $p = 0.005$). Significant differences in the two groups were also the COPD (32.2% vs. 18.7%; $p = 0.031$), neurological syndrome (57.1% vs. 41.3%, $p = 0.035$), autonomy

**Table 2. Comparison of characteristics of 264 people living in retirement nursing homes with SARS-CoV-2 infection, divided into people with (132) and without (132) SARS-CoV-2 related symptoms.**

|  | Total (264) | Symptomatic (132) | Asymptomatic (132) | *p-value* |
|---|---|---|---|---|
| Age, year (mean ± SD) | 81.9 ± 10.1 | 82.3 ± 9.7 | 81.5 ± 10.5 | 0.52 |
| **Gender, n(%)** |  |  |  |  |
| Male | 99 (37.5) | 54 (40.9) | 45 (34.1) | 0.253 |
| **Comorbidities, n(%)** |  |  |  |  |
| BMI > 30 | 53 (20.1) | 29 (22) | 24 (18.2) | 0.461 |
| Hypertension | 159 (60.2) | 86 (65.2) | 74 (56.1) | 0.131 |
| Diabetes | 55 (20.8) | 24 (18.2) | 31 (23.5) | 0.366 |
| COPD | 57 (21.6) | 37 (28) | 20 (15.2) | 0.011 |
| CHD | 99 (37.5) | 58 (43.9) | 41 (31.1) | 0.031 |
| Mental disorders | 126 (47.7) | 61 (46.2) | 65 (49.2) | 0.622 |
| Neurological syndromes | 117 (44.3) | 69 (52.3) | 49 (37.1) | 0.013 |
| Kidney failure | 20 (7.6) | 13 (9.8) | 7 (5.3) | 0.163 |
| Malignancy | 26 (9.8) | 12 (9.1) | 14 (10.6) | 0.6 |
| Hypokinetic disease | 70 (26.5) | 41 (31.1) | 30 (22.7) | 0.127 |
| **Chronic treatment, n(%)** |  |  |  |  |
| ARBs | 45 (17) | 28 (21.2) | 17 (12.9) | 0.072 |
| ACE | 57 (21.6) | 29 (22) | 28 (21.2) | 0.881 |
| NOACs | 34 (12.9) | 18 (13.6) | 16 (12.1) | 0.713 |
| VKAs | 14 (5.3) | 7 (5.3) | 7 (5.3) | 1 |
| LMWH | 23 (8.7) | 15 (11.4) | 8 (6.1) | 0.131 |
| **Death, n(%)** | 56 (21.2) | 37 (28) | 19 (14.4) | 0.007 |

* Calculated with chi-squared test or t-test as appropriate. SD: standard deviation; BMI: body mass index; COPD: chronic obstructive pulmonary disease; CHD: cardiovascular disease; ARBs: Angiotensin II receptor blockers; ACE: angiotensin-converting enzyme. NOACs: Non-vitamin K oral anticoagulants; VKAs: Vitamin K antagonists; LMWH: low molecular weight heparin.

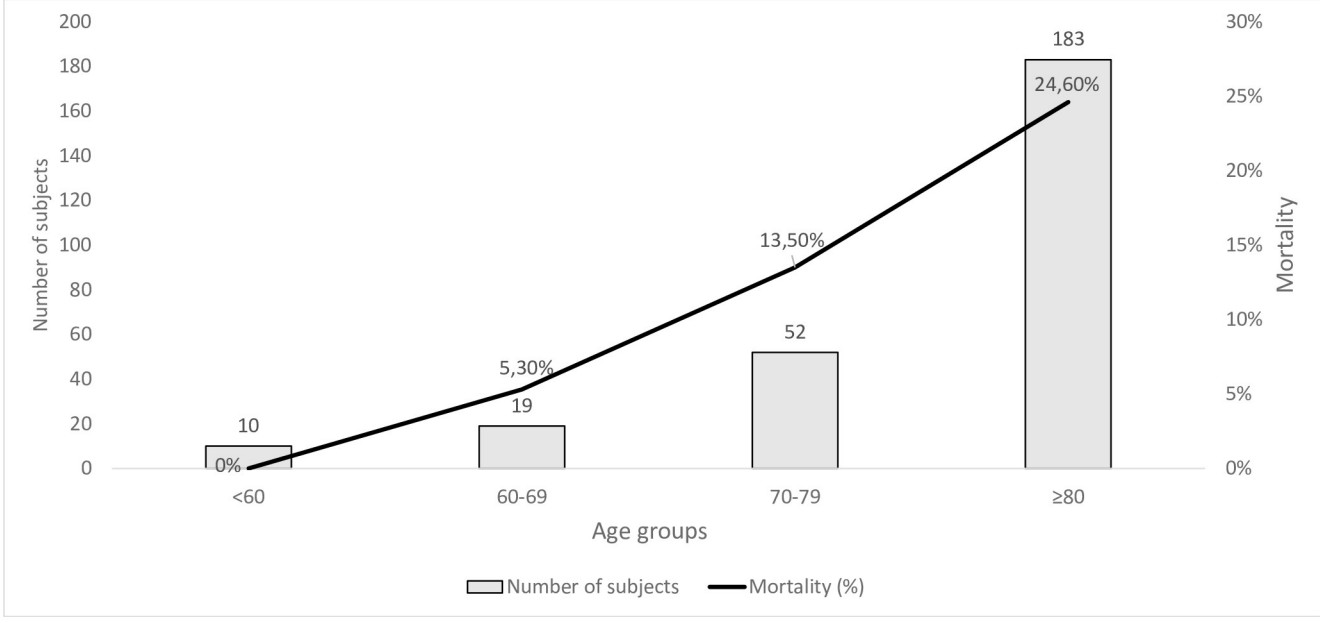

**Fig 2. Distribution of 264 patients with SARS-CoV-2 infection according to age groups and mortality rates.**

**Table 3. Comparison of characteristics of 264 people living in retirement nursing homes with SARS-CoV-2 infection, divided into deceased and survival.**

|  | Total (264) | Deceased (56) | Survival (208) | *p-value* |
|---|---|---|---|---|
| Age, year (mean ± SD) | 81.9 ± 10.1 | 85.3 ± 6.3 | 81 ± 10.7 | 0.005 |
| **Gender, n(%)** |  |  |  |  |
| Male | 99 (37.5) | 26 (46.4) | 73 (35.1) | 0.12 |
| **Comorbidity, n(%)** |  |  |  |  |
| BMI > 30 | 53 (20.1) | 12 (21.4) | 41 (19.7) | 0.729 |
| Hypertension | 159 (60.2) | 29 (51.8) | 131 (63) | 0.128 |
| Diabetes | 55 (20.8) | 14 (25) | 42 (21.2) | 0.435 |
| COPD | 57 (21.9) | 18 (32.1) | 39 (18.7) | 0.031 |
| CHD | 99 (37.5) | 23 (41.1) | 76 (36.5) | 0.534 |
| Mental disorders | 126 (47.7) | 23 (41.1) | 103 (49.5) | 0.261 |
| Neurological syndromes | 117 (44.3) | 32 (57.1) | 86 (41.3) | 0.035 |
| Kidney failure | 20 (7.6) | 5 (8.9) | 15 (7.2) | 0.666 |
| Malignancy | 26 (9.8) | 4 (7.1) | 22 (1.6) | 0.444 |
| Autonomy | 83 (31.4) | 2 (3.6) | 81 (38.9) | <0.001 |
| Hypokinetic disease | 70 2(6.5) | 31 (55.4) | 40 (19.2) | <0.001 |
| **Symptomatology, n(%)** |  |  |  |  |
| Fever + Dyspnea | 55 (20.8) | 24 (42.9) | 31 (14.9) | <0.001 |
| **Chronic treatment, n(%)** |  |  |  |  |
| ARBs | 45 (17) | 11 (19.6) | 34 (16.3) | 0.56 |
| ACE | 57 (21.6) | 10 (17.9) | 47 (22.6) | 0.444 |
| **SARS-COV-2 Treatment, n(%)** |  |  |  |  |
| Hydroxychloroquine | 59 (22.3) | 12 (21.4) | 47 (22.6) | 0.954 |
| Azithromycin | 48 (18.2) | 9 (16.1) | 39 (18.7) | 0.800 |
| LMWH | 105 (39.8) | 16 (28.6) | 89 (42.8) | 0.111 |

* Calculated with chi-squared test or t-test as appropriate. SD: standard deviation; BMI: body mass index; COPD: chronic obstructive pulmonary disease; CHD: cardiovascular disease; ARBs: Angiotensin II receptor blockers; ACE: angiotensin-converting enzyme; LMWH: low molecular weight heparin.

(3.6% vs. 38.9%, *p*<0.001), hypokinetic disease (55.4% vs. 19.2%), and presence of fever and dyspnea (42.9% vs. 14.9%, *p*<0.001) (Table 3).

At the multivariate Cox proportional-hazards model (S3 Table) the significant factors associated with 30-days mortality were male gender [Hazard Ratio (HR) 1.93 (95%CI 1.07–3.45), *p* = 0.028], hypokinetic disease [HR 1.91 (95%CI 1.05–3.48), *p* = 0.035], and the presence of fever and dyspnea [HR 3.99 (95%CI 2.05–7.79), *p*<0.001]. On the contrary, patients' autonomy [HR 0.051 (95%CI 0.007–0.39), p = 0.004], and heparin treatment [HR 0.42 (95%CI 0.22–0.79), p = 0.008], were associated with a significantly lower mortality risk, according to our results' (Fig 3).

## Discussion

To our knowledge, this report is one of the largest studies about the SARS-CoV-2 spread in retirement nursing homes.

Nursing homes play an important role as a significant clustering hotspot of the epidemic, as reported in the literature [26, 27]. However, few published official data regarding the infection rate in nursing homes, individual risk factors, and the residents' outcome. For this reason, focused interventions should be realized in these settings.

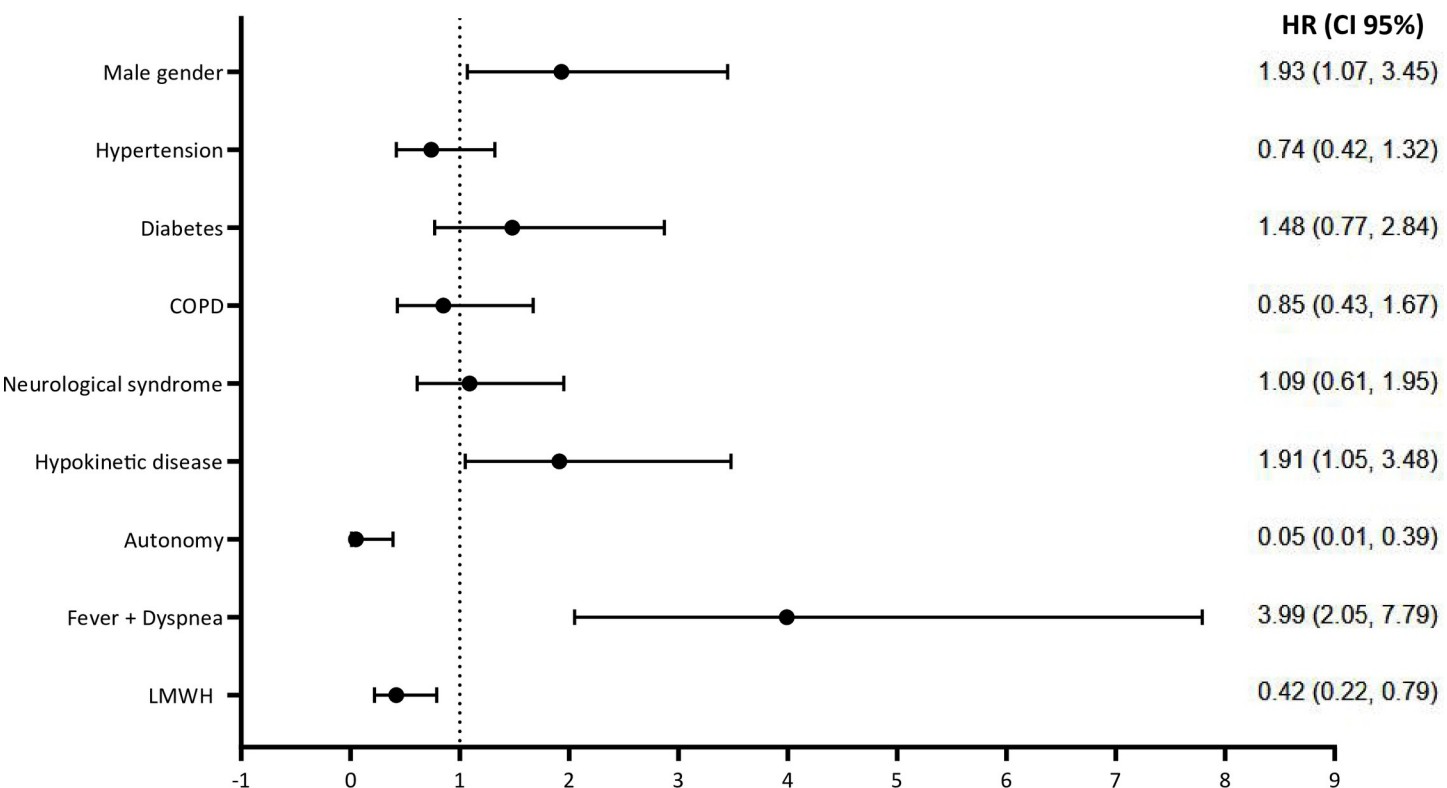

**Fig 3. Forrest Plots of multivariate Cox proportional hazards model regression estimates of factors associated with mortality.** Multivariable model includes all variables selected by a backward selection that were retained with a p-value less than 0.2 level. HR: Hazard Ratio; CI: Confidence Interval; COPD: chronic obstructive pulmonary disease; LMWH: low molecular weight heparin.

In the facilities with at least one SARS-CoV-2 positive subject, 69.1% of the PLRNH resulted positive in our study, thus, confirming that retirement nursing homes have high transmission rates of SARS-CoV-2. We can hypothesize that this could be related to crowding, sharing of gathering areas, and inadequate infection prevention and control measures, which need to be quickly revised [28, 29]. One of the most critical issues in the authors' experience was applying optimal protocols of isolation and social distancing in such challenging settings and difficulties in communication with elderly subjects, often with mental health and neurological disorders. Furthermore, as shown by Daoust, elderly people seem to have reduced adherence to preventive measures (e.g., wearing masks and social distancing) [30].

Wang et al., in their manuscript among 339 elderly patients with SARS-CoV-2 infection, found that the most common comorbidities were hypertension (40.8%), diabetes (16.0%), and CHD (15.7%) [31]. In our study, the percentage of people with these comorbidities was abundantly higher, and other comorbidities such as neurological syndromes and mental disorders were more common than diabetes and CHD. The older age of our cases could explain this difference (81.2±10.1 in our study vs. 71±8).

When considering factors that influenced infection occurrence, mental health disorders, malignancies, and use of ARBs were significantly associated with SARS-CoV-2 infection development at multivariate analysis.

Regarding ACE inhibitors, the interaction between SARS viruses and ACE 2 has been proposed as a potential risk factor of infectivity. For this reason, some concerns have been raised about their use, for a possible transmission increase, related to hosting susceptibility.

Interestingly, our findings diverge from most common literature findings, and ACE inhibitors did not influence the risk, as described by most of the literature on SARS-CoV-2 infection and outcomes [32–36].

On the contrary, our data seem to support the hypothesis that chronic treatment with ARBs can raise the possibility of contracting the disease.

Regarding asymptomatic patients, estimates among the elderly are debated. A recent meta-analysis, including 50,155 subjects, estimated the prevalence of asymptomatic SARS-CoV-2 infection between 15 and 16%. When analysing the elderly subgroup, the prevalence of asymptomatic patients was 28.3% [37]. Our data showed an even higher prevalence, with the involvement of 50% of PLRNH. The overestimation of asymptomatic cases could explain this high percentage. In fact, it was difficult to collect medical history regarding slight symptoms, giving the high presence of neurological syndromes, which represents a barrier between doctor and patient.

Mildly symptomatic and pre-symptomatic cases can be incorrectly diagnosed as asymptomatic, as showed by previously published data on long-term care facilities [38]. In any case, a high proportion of asymptomatic, mildly symptomatic, and pre-symptomatic patients support a possible important role of unrecognized carriers in SARS-CoV-2 diffusion and transmission in retirement nursing homes [38–40].

At the multivariate logistic regression, COPD and neurological syndromes were associated with a higher risk of developing SARS-CoV-2-related symptoms. On the one hand, the ACE2 upregulation may have a protective effect on the lung epithelial cells against the chronic injury observed in smokers and COPD patients. On the other hand, it could be related to SARS-CoV-2 mucosal penetration, with a significant role in the clinical presentation [41].

Association between neurological syndromes and COVID-19 development is still debated. Stoessl et al. and Ferini-Strambi et al. aimed that Parkinson's disease and parkinsonism are not related to a major infection vulnerability [42, 43]. However, old age and respiratory muscle rigidity associated with neurological disorders, as well as the presence of comorbidities, could explain our results.

To our knowledge, there are no data available about the risk of developing COVID-19 related symptoms in patients with dementia and Alzheimer's disease. It is known how dementia, especially in its severe stages and independently by age, is associated with worse outcomes [13, 44]. Probably, elderly patients' vulnerability may be associated with a more critical presentation of COVID-19.

The most common symptoms were fever and dyspnoea. As highlighted by other studies, fever represents the most common symptom among elderly patients. From this point of view, our data are concordant with the literature. However, cough, dyspnoea, or sputum have been reported with different frequencies in previous reports, suggesting that these symptoms may be extremely variable as presentation, compared to fever [14, 31, 45].

Regarding mortality, in our sample, it occurred both in symptomatic and asymptomatic patients. Overall, 56 (21.2%) people with SARS-CoV-2 died, while the negative group percentage was 11%.

Bonadad et al.[46] conducted a meta-analysis on 611,583 subjects to analyze the effect of age on mortality. Comparing the different samples, the mortality was lower in our study for all age groups (60–69 years: 5.3% vs. 9.5%; 70–79 years: 13.5% vs. 22.8%; ≥80 years: 24.6% vs. 29.6%).

The role of the male gender on mortality is still debated. Looking at the national report, in 54/64 (84%) countries, the male/female death ratio was>1 [47]. Data from different Countries highlighted a fatality ratio higher among men than in women. Although some authors

explained this difference with higher rates of comorbidities among men [48], prior studies suggested a higher trend among females to follow hand hygiene [49] and preventive care [48, 50].

The hypokinetic disease has been identified as a mortality predictor, with an HR of 1.91. It could be explained by the fact that subjects with this condition are, generally, more vulnerable.

Even if having a malignancy was not associated with an increased risk of symptoms' development and mortality, these results should be taken with caution, given the low number of cases.

Patients who started treatment with LMWH showed decreased mortality. The SARS-CoV-2 pathogenesis is still unclear. However, many reports associated COVID-19 and coagulopathies, such as pulmonary embolism and microvascular thrombosis [16, 51]. For these reasons, LMWH is used in the management of COVID-19 [52]. Different studies showed how LMWH has anti-inflammatory and, possibly, antiviral effects [53–55]. In our study, 19 asymptomatic patients died suddenly. Post-mortem examinations were not performed, but the presence of coagulopathy was likely. Furthermore, different studies described the presence of undiagnosed cerebrovascular and cardiovascular complications in severe SARS-CoV-2 infections [56, 57]. We speculate that LMWH start could also be considered in asymptomatic patients, based on the patients' comorbidities and the fall-risk evaluation.

The receipt of treatment with hydroxychloroquine and azithromycin did not modify the mortality, in accordance with previous studies [18, 23, 24].

The COVID-19 may have relatively high mortality rates among elderly patients. For this reason, focused interventions should be programmed by healthcare providers, with the provision of close monitoring in patients at high mortality risk due to their burden of comorbidities. More studies are needed to evaluate the advantages in the survival of an early LMWH start, particularly in such a challenging and frail population.

### Limitations of the study

Our study has some limitations that should be addressed. Firstly, this is an observational, retrospective survey. Secondly, our experience is monocentric, and this could not entirely reflect the national situation. Given the timeline and the constant advice updates, treatments were extremely variable in time of staring after symptoms development, drug choice, and combination. This could cause a bias in our evaluation of hydroxychloroquine or azithromycin influence on mortality, as well as severe comorbidities presence.

### Supporting information

**S1 Table. Bivariate and multivariate logistic regression estimates of factors associated with SARS-CoV-2 infection.**
(DOCX)

**S2 Table. Bivariate and multivariate logistic regression estimates of factors associated with SARS-CoV-2-related symptoms.**
(DOCX)

**S3 Table. Bivariate and multivariate Cox proportional-hazards model estimates of factors associated with mortality.**
(DOCX)

**S1 Fig. Forrest Plots of multivariate logistic regression estimates of factors associated with symptoms development.** Multivariable model includes all variables selected by a backward selection that were retained with a p-value less than 0.2 level. OR: Odds Ratio; CI: Confidence Interval; COPD: chronic obstructive pulmonary disease; CHD: cardiovascular disease; ARBs:

Angiotensin II receptor blockers.
(PDF)

**S1 File.**
(XLSX)

**S2 File.**
(XLSX)

## Author Contributions

**Conceptualization:** Andrea De Vito, Elija Princic, Nicholas Geremia, Giordano Madeddu, Sergio Babudieri.

**Data curation:** Andrea De Vito, Elija Princic, Nicholas Geremia, Catello Mario Panu Napodano, Alberto Augusto Muredda.

**Formal analysis:** Andrea De Vito, Vito Fiore.

**Investigation:** Andrea De Vito, Elija Princic, Nicholas Geremia, Catello Mario Panu Napodano, Alberto Augusto Muredda, Ivana Maida.

**Methodology:** Vito Fiore.

**Supervision:** Ivana Maida, Giordano Madeddu, Sergio Babudieri.

**Validation:** Giordano Madeddu, Sergio Babudieri.

**Writing – original draft:** Andrea De Vito, Vito Fiore, Elija Princic, Nicholas Geremia, Giordano Madeddu.

**Writing – review & editing:** Andrea De Vito, Vito Fiore, Catello Mario Panu Napodano, Alberto Augusto Muredda, Ivana Maida, Giordano Madeddu, Sergio Babudieri.

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
