## [Decision Letter · Decision Letter 0]

6 Jan 2021

PONE-D-20-34816

Predictors of infection, symptoms’ development, and mortality in people with SARS-CoV-2 living in retirement nursing homes.

PLOS ONE

Dear Dr. De Vito,

Thank you for submitting your manuscript to PLOS ONE. After careful consideration, we feel that it has merit but does not fully meet PLOS ONE’s publication criteria as it currently stands. Therefore, we invite you to submit a revised version of the manuscript that addresses the points raised during the review process.

We look forward to receiving your revised manuscript.

Kind regards,

Andrea Calcagno

Academic Editor

PLOS ONE

Journal Requirements:

2. In the Methods section and in the online submission please specify the name of the local IRB institute which approved the study and please include a copy of the original ethics documents and an English translated version as "Supporting" file.

Additional Editor Comments (if provided):

Dear authors,

I have received the comments of 2 reviewers and they suggest major revision.

I agree with them that some caution may be used in interpreting the data and that some may be further discussed.

I also agree that hydroxycholoroquine data should be taken with caution given the study design and the results from randomized studies.

Reviewers' comments:

Reviewer's Responses to Questions

**Comments to the Author**

1. Is the manuscript technically sound, and do the data support the conclusions?

Reviewer #1: No

Reviewer #2: Yes

2. Has the statistical analysis been performed appropriately and rigorously? 

Reviewer #1: Yes

Reviewer #2: Yes

3. Have the authors made all data underlying the findings in their manuscript fully available?

Reviewer #1: No

Reviewer #2: Yes

4. Is the manuscript presented in an intelligible fashion and written in standard English?

Reviewer #1: No

Reviewer #2: Yes

5. Review Comments to the Author

Reviewer #1: This paper reports on rates of SARS-CoV-2 positivity in ‘retirement nursing homes’ experiencing outbreaks in Italy during April/May 2020 and comments on symptoms as well as clinical outcomes. The study also looks at the associations of different putative treatments and their relationship to outcomes. A substantial number (382) of patients are included in the study. This is an important topic and looks at a critically high risk group which is not easy to study.

Unfortunately however there is sufficient detail in the methods for me to be able to recommend this paper for publication, and I am concerned that the conclusions, particularly with respect to recommending treatments, stray to far from the limitations of the data.

Specific comments:

1. The patient population is not well defined. Although this may be an English language translation issue. The entity referred to in the manuscript is ‘retirement nursing homes’. It is unclear if this refers to people in sheltered care/ warden controled accommodation who require very little support; or residential Care home residents (requiring support with some activities of daily living) or nursing home residents (requiring nursing care specifically, ie. a high degree of dependency).

2. I am not convinced that the background literature review is sufficiently rigorous / critical. For example the paper implies that there is ‘debate’ around the efficacy of hydroxychloroquine (HCQ). On the contrary there are plenty of very well designed randomised trials which do not show efficacy and I am concerned that framing this as debate could give the wrong impression.

3. The description of the study could and should be much clearer. This would allow readers to interpret the findings in the right context. Given that the authors wish to comment on the efficacy of putative treatments should this be framed as a ‘retrospective non-randomised non-blinded cohort study’?

4. It is unclear who was given which treatment, at what dose, and why.

5. The testing methodology is unclear. Was this one off? What prompted testing? What was the timing? Were people tested repeatedly?

6. How were symptoms ascertained? Were patients who had no symptoms reassessed to see if they were truly asymptomatic, vs pre-symptomatic? The authors comment on clinical symptoms and signs “at admission”. Does this mean the patient were hospiasllied ? The rest of the paper makes this sound as though patients were managed in the community. This would need clarification. It is suggested that several asymptomatic patients died - did they really develop no symptoms prior to death, even in their final hours?

7. Were there any adverse events related to treatment, such as but not limited to bleeding or QTc elongation ?

8. Are comparisons corrected for multiple comparisons? The full range of possible comorbidities examined is confusing. In earlier tables neurological syndromes are lumped together and later hypokinetic disease appears separately.

9. The interpretation of these findings is generous. It is probably not appropriate to say ‘protective factors were’ in the results section. All that can be commented on was an association (protection suggests causality). I do not regard this as sufficient evidence to support giving LMWH to patients in this setting and I think the authors should be careful about doing so, or make a clearer justification of why they think this should be done.

Reviewer #2: This is an interesting research on a timely and important topic. While I cannot recommend the publication of the research as of now, I would like to give the authors a chance to respond to my comments below.

General. I don’t see any reason to explain why the authors are so concise when they describe their measures. For example, patient compliance is defined on lines 87-89, but it is not obvious that it is related to general medical advices given that the reader is aiming to read a very specific study on COVID-19. At first, I understood that compliance was about the COVID-19-related treatments. Hence, the results for those who did not contract COVID-19 did not make much sense. Another example: mental disorders, which is labelled as psychiatric disorders in the text. What does it mean? It can be very brief, but a little bit of details would be appreciated.

There is something incredibly interesting about autonomy and mortality rates. Among 83 observations with a value on ‘autonomy’, 81/83 survived. This is probably the most interesting result of the study, but it is not discussed. First, and again, the variable should be detailed in greater length. What does it mean (‘any daily activity’ is not that obvious)? Second, what are the implications? Is it simply a confounding factor measuring something else? It is *very* a strong predictor of surviving, which is very important, so what should we make out of this finding?

Statistical analysis section. First, the mention of ‘univariate analysis’ seems wrong: it’s always bivariate or multivariate. For example, the variables are break down by positivity/negativity (Table 1), Symptomatic or not (Table 2) and death/survival (Table 3). That’s not univariate. Second, there is no need to say anything about the ‘established p-values’ as the authors show the exact p-value instead of using asterix (which is the right thing to do). However, there is a mention that only the variable where p<.2 where included. This is ad hoc and needs a rationale. I am still unsure what it means and what happened, and it should be justified.

On the data: we have no idea about the location. I appreciate ethical concerns, but it seems that mentioning the country would be the minimum. It makes little sense that it is absent from the manuscript, but I failed to find such important. There is a mention of a ‘province…’ I might have missed something here.

The findings for cancer entail way too uncertainty to be taken seriously. The number of observations is too low (26) and the distributions make it worse (12 v/s 14 in Table 2; 4 v/s 22 in Table 3).

The authors should mention the studies by Daoust (2020) in PLOS One about elderly people and their response to COVID-19. Among others, it is relevant for the discussion section.

Line 160: the comparison for the effect of age is ad hoc, using the cut-off of 80 years with no justification. There is no need for such a dichotomization.

On hydroxychloroquine: The authors cannot claim that it is ‘probably due to the fact that the subjects that started the treatments had severe disease’ as it would be misleading given what we now know about this treatment. It seems simply not effective. See, among others, the meta-analysis and systematic review by Fiolet et al. (2020, in CMI).

6. PLOS authors have the option to publish the peer review history of their article (what does this mean?). If published, this will include your full peer review and any attached files.

Reviewer #1: No

Reviewer #2: No

---

## [Author Response · Author response to Decision Letter 0]

28 Jan 2021

Sassari, 20/01/2021

Dear Professor Calcagno,

Academic Editor

PLOS ONE

I would like to thank you for your interest in our manuscript and the possibility of revising our work. On behalf of my co-authors, I submit the revised version of 'Predictors of infection, symptoms' development, and mortality in people with SARS-CoV-2 living in retirement nursing homes'.

We would also thank the reviewers for helping us improving our manuscript quality through the evaluable comments received. We have highlighted in yellow the changes to the manuscript. Furthermore, rereading the manuscript, we found some typos that we have fixed.

Please, find below the point-by-point author response.

Reviewer (R) #1:

1. The patient population is not well defined. Although this may be an English language translation issue. The entity referred to in the manuscript is 'retirement nursing homes'. It is unclear if this refers to people in sheltered care/ warden controled accommodation who require very little support; or residential Care home residents (requiring support with some activities of daily living) or nursing home residents (requiring nursing care specifically, ie. a high degree of dependency).

Authors' response (AR): Thank you for your valuable comment. Effectively, there is an important translation issue. There is no clear difference between sheltered care accommodations, residential care homes, and nursing homes in Italian settings. Therefore, the population considered in our study included people requiring low-level support, support in some activities, or with high dependence degree. In all nursing homes included in the study, nurses and assistive personnel were present 24 hours a day 7 days a week, whereas primary care doctors, and other specialists, were present only upon request. We added a 'patients definition' subparagraph to explain the definitions better, and the population included.

2. I am not convinced that the background literature review is sufficiently rigorous / critical. For example the paper implies that there is 'debate' around the efficacy of hydroxychloroquine (HCQ). On the contrary there are plenty of very well designed randomised trials which do not show efficacy and I am concerned that framing this as debate could give the wrong impression.

AR: Thank you for your comment. We agree with you that there are many trials in which HCQ showed to be ineffective. However, some other well-designed studies and meta-analyses evidenced the efficacy of HCQ. (Million et al., CMI 2020 https://doi.org/10.1016/j.nmni.2020.100709; Prodromos et al. NMNI 2020 https://doi.org/10.1016/j.nmni.2020.100776; D'Arminio Manforte et al. IJID 2020 https://doi.org/10.1016/j.ijid.2020.07.056, Todd et al. IJID https://doi.org/10.1016/j.ijid.2020.06.095 ). We have discussed this aspect better in the introduction, adding that more than 30 countries do not recommend using HCQ in clinical practice.

3. The description of the study could and should be much clearer. This would allow readers to interpret the findings in the right context. Given that the authors wish to comment on the efficacy of putative treatments should this be framed as a 'retrospective non-randomised non-blinded cohort study'?

AR: Thank you for your comment. According to the literature, we defined it as an observational retrospective cohort study. 

4. It is unclear who was given which treatment, at what dose, and why.

AR: SARS-COV-2 infection treatments consisted of hydroxychloroquine 200 mg twice daily, azithromycin 500 mg once daily for five days, or LMWH adjusted by body weight, kidney failure, and HAS-BLED score. Treatment was started only in symptomatic patients. According to the time of enrolment, the treatment prescription was based on national guidelines, and it was adjusted according to the disease's clinical presentation and drug-drug interactions with patients' chronic therapy. Furthermore, we decided to avoid HCQ and azithromycin in patients with a previous diagnosis of cardiovascular disease, in people with G6PDH deficiency, and those who had a previous side effect to one of these drugs. We specified this aspect better in the method's section.

5. The testing methodology is unclear. Was this one off? What prompted testing? What was the timing? Were people tested repeatedly?

AR: Thank you for your comment. Since the start of the outbreak, in our province, a total of 1946 subjects living in 63 different retirement nursing homes were tested for SARS-CoV-2. Nasopharyngeal swabs were repeated in positive patients every week until the first negative result and every week until the end of the outbreak among people who tested negative. We added a paragraph in the Methods section.

6. How were symptoms ascertained? Were patients who had no symptoms reassessed to see if they were truly asymptomatic, vs pre-symptomatic? The authors comment on clinical symptoms and signs "at admission". Does this mean the patient were hospiasllied ? The rest of the paper makes this sound as though patients were managed in the community. This would need clarification. It is suggested that several asymptomatic patients died - did they really develop no symptoms prior to death, even in their final hours?

AR: Thank you for giving us the possibility to clarify this point better. 

- Patients' history was collected by the available medical records. Medical assessment was conducted with clinical evaluation every three days, and electrocardiography at time zero and day three of treatment, with a portable device. Blood exams were performed according to the physician's request. Nasopharyngeal swab was repeated every week till a negative result among positive patients and till the end of the epidemiological focus among people who tested negative. We added a specific 'Medical assessment' subparagraph in the Results section if it could be appreciated.

- Patients were managed in the community. We apologize for the mistake. We changed 'at admission' with 'at first evaluation'.

- All asymptomatic patients who died during the period of study did not show any COVID-19 symptom. 

7. Were there any adverse events related to treatment, such as but not limited to bleeding or QTc elongation ?

AR: Thank you for your comment. We did not find any QT prolongation due to the treatment. However, we decided not to administered HCQ and/or azithromycin in people with known cardiological diseases. We added a specific comment in the methods section. 

8. Are comparisons corrected for multiple comparisons? The full range of possible comorbidities examined is confusing. In earlier tables neurological syndromes are lumped together and later hypokinetic disease appears separately.

AR: With hypokinetic disease, we would indicate all people who were forced to stay in bed for several conditions, but we agree with you that it could create a misunderstanding. We decided to substitute hypokinetic disease with bedridden. We have added in the methods section the definition of bedridden. 

9. The interpretation of these findings is generous. It is probably not appropriate to say 'protective factors were' in the results section. All that can be commented on was an association (protection suggests causality). I do not regard this as sufficient evidence to support giving LMWH to patients in this setting and I think the authors should be careful about doing so, or make a clearer justification of why they think this should be done.

AR: Thank you for your precious comment. If it could be appreciated, we rephrased as 'patients' autonomy [HR 0.051 (95%CI 0.007-0.39), p=0.004], and heparin treatment [HR 0.42 (95%CI 0.22-0.79), p=0.008], were associated with a significantly lower mortality risk, according to our results'. 

Reviewer #2 

1. General. I don't see any reason to explain why the authors are so concise when they describe their measures. For example, patient compliance is defined on lines 87-89, but it is not obvious that it is related to general medical advices given that the reader is aiming to read a very specific study on COVID-19. At first, I understood that compliance was about the COVID-19-related treatments. Hence, the results for those who did not contract COVID-19 did not make much sense. Another example: mental disorders, which is labelled as psychiatric disorders in the text. What does it mean? It can be very brief, but a little bit of details would be appreciated.

AR: Thank you for your comment. Since our study included both SARS-CoV-2 infected and not infected people, we referred to general terms when explained patient compliance and autonomy. About psychiatric/mental disorders, we agree with you that the use of different terms could be confusing. We provided to level out the manuscript always using "mental disorders".

2. There is something incredibly interesting about autonomy and mortality rates. Among 83 observations with a value on 'autonomy', 81/83 survived. This is probably the most interesting result of the study, but it is not discussed. First, and again, the variable should be detailed in greater length. What does it mean ('any daily activity' is not that obvious)? Second, what are the implications? Is it simply a confounding factor measuring something else? It is *very* a strong predictor of surviving, which is very important, so what should we make out of this finding?

AR: Thank you for your useful comment. We better explained what we meant as autonomy. As further specified in the methods section, we considered autonomous all people with an ADL score of 5 or 6. People with 5 or 6 ADL scores probably had a better health status, explaining the lower mortality. 

3. Statistical analysis section. First, the mention of 'univariate analysis' seems wrong: it's always bivariate or multivariate. For example, the variables are break down by positivity/negativity (Table 1), Symptomatic or not (Table 2) and death/survival (Table 3). That's not univariate. Second, there is no need to say anything about the 'established p-values' as the authors show the exact p-value instead of using asterix (which is the right thing to do). However, there is a mention that only the variable where p<.2 where included. This is ad hoc and needs a rationale. I am still unsure what it means and what happened, and it should be justified.

AR: Thank you for your comment. We changed the word 'univariate' with 'bivariate'. We deleted the established significance level accordingly. Variable with p<0.2 in the bivariate, were included in the multivariate logistic regression. We better clarified it in the text.

4. On the data: we have no idea about the location. I appreciate ethical concerns, but it seems that mentioning the country would be the minimum. It makes little sense that it is absent from the manuscript, but I failed to find such important. There is a mention of a 'province…' I might have missed something here.

AR: We added the city and country, according to your suggestion.

5. The findings for cancer entail way too uncertainty to be taken seriously. The number of observations is too low (26) and the distributions make it worse (12 v/s 14 in Table 2; 4 v/s 22 in Table 3).

AR: We agree with you about the low number of malignancy cases. Therefore, we added a sentence (Even if having a malignancy was not associated with an increased risk of symptoms' development and mortality, these results should be taken with caution, given the low number of cases.) in the discussion section. 

6. The authors should mention the studies by Daoust (2020) in PLOS One about elderly people and their response to COVID-19. Among others, it is relevant for the discussion section.

AR: thank you for your suggestion. We provided to discuss this interesting article as you suggested (line 228).

7. Line 160: the comparison for the effect of age is ad hoc, using the cut-off of 80 years with no justification. There is no need for such a dichotomization.

AR: Thank you for your comment. We want to specify that we did not use a dichotomization. We divided patients by 10 years periods (<60, 60-69, 70-79 and >80) as shown in Figure 2. Furthermore, we observed the mortality between 80/89 and 90-99 did not differ significantly. 

8. On hydroxychloroquine: The authors cannot claim that it is 'probably due to the fact that the subjects that started the treatments had severe disease' as it would be misleading given what we now know about this treatment. It seems simply not effective. See, among others, the meta-analysis and systematic review by Fiolet et al. (2020, in CMI).

AR: Thank you for your comment. In the Discussion section, we modified the sentence accordingly.

---

## [Decision Letter · Decision Letter 1]

9 Feb 2021

PONE-D-20-34816R1

Predictors of infection, symptoms’ development, and mortality in people with SARS-CoV-2 living in retirement nursing homes.

PLOS ONE

Dear Dr. De Vito,

Thank you for submitting your manuscript to PLOS ONE. After careful consideration, we feel that it has merit but does not fully meet PLOS ONE’s publication criteria as it currently stands. Therefore, we invite you to submit a revised version of the manuscript that addresses the points raised during the review process.

We look forward to receiving your revised manuscript.

Kind regards,

Andrea Calcagno

Academic Editor

PLOS ONE

Additional Editor Comments (if provided):

Thanks for addressing all the reviewers's comments. I believe the manuscript has singifiafcntly improved. One of the reviewer has still a question about p values that I ask you to address before accepting it for publication.

Reviewers' comments:

Reviewer's Responses to Questions

**Comments to the Author**

1. If the authors have adequately addressed your comments raised in a previous round of review and you feel that this manuscript is now acceptable for publication, you may indicate that here to bypass the “Comments to the Author” section, enter your conflict of interest statement in the “Confidential to Editor” section, and submit your "Accept" recommendation.

Reviewer #2: All comments have been addressed

2. Is the manuscript technically sound, and do the data support the conclusions?

Reviewer #2: Yes

3. Has the statistical analysis been performed appropriately and rigorously? 

Reviewer #2: Yes

4. Have the authors made all data underlying the findings in their manuscript fully available?

Reviewer #2: Yes

5. Is the manuscript presented in an intelligible fashion and written in standard English?

Reviewer #2: Yes

6. Review Comments to the Author

Reviewer #2: The authors have addressed all my concerns except one, that is, the second consideration of my third point: "Second, there is no need to say anything about the

'established p-values' as the authors show the exact p-value instead of using asterix

(which is the right thing to do). However, there is a mention that only the variable where

p<.2 where included. This is ad hoc and needs a rationale. I am still unsure what it

means and what happened, and it should be justified."

I am still puzzled after reading the authors' responses and the new version of the manuscript.

7. PLOS authors have the option to publish the peer review history of their article (what does this mean?). If published, this will include your full peer review and any attached files.

Reviewer #2: No

---

## [Author Response · Author response to Decision Letter 1]

11 Feb 2021

Dear Professor Calcagno,

I would like to thank you for allowing us to reply to reviewer n°2.

Reviewer concern: 

Second, there is no need to say anything about the 'established p-values' as the authors show the exact p-value instead of using Asterix (which is the right thing to do). 

However, there is a mention that only the variable where p<.2 where included. This is ad hoc and needs a rationale. I am still unsure what it means and what happened, and it should be justified."

Authors reply:

Thank you for your comment. We included all variables in the multivariate model with a p-value <0.2 because we preferred to be conservative. Otherwise, we could have decided to use a less conservative model, including only variables with a p-value <0.1 or 0.15, but this way, there could have been the possibility to exclude some clinically important variables or adjustments.

Regarding the choice to specify that we considered the p-value statistically significant when <0.05, we preferred to do so to make it as clear as possible to the reader. 

Yours sincerely,

Andrea De Vito

---

## [Editor Report · Decision Letter 2]

18 Feb 2021

Predictors of infection, symptoms’ development, and mortality in people with SARS-CoV-2 living in retirement nursing homes.

PONE-D-20-34816R2

Dear Dr. De Vito,

We’re pleased to inform you that your manuscript has been judged scientifically suitable for publication and will be formally accepted for publication once it meets all outstanding technical requirements.

Kind regards,

Andrea Calcagno

Academic Editor

PLOS ONE

Additional Editor Comments (optional):

Thanks for addressing the last concerns from one of our reviewers. I think the manuscript can now be accepted for publication.
---

## [Editor Report · Acceptance letter]

8 Mar 2021

PONE-D-20-34816R2 

Predictors of infection, symptoms development, and mortality in people with SARS-CoV-2 living in retirement nursing homes. 

Dear Dr. De Vito:

I'm pleased to inform you that your manuscript has been deemed suitable for publication in PLOS ONE. Congratulations! Your manuscript is now with our production department. 

Kind regards, 

on behalf of

Dr. Andrea Calcagno 

Academic Editor

PLOS ONE